# Cardiac Arrest Occurring in High-Rise Buildings: A Scoping Review

**DOI:** 10.3390/jcm10204684

**Published:** 2021-10-13

**Authors:** Ming Xuan Han, Amelia Natasha Wen Ting Yeo, Marcus Eng Hock Ong, Karen Smith, Yu Liang Lim, Norman Huangyu Lin, Bobo Tan, Shalini Arulanandam, Andrew Fu Wah Ho, Qin Xiang Ng

**Affiliations:** 1Emergency Medical Services Department, Singapore Civil Defence Force, Singapore 408827, Singapore; mxhan9598@yahoo.com (M.X.H.); lim_yu_liang@scdf.gov.sg (Y.L.L.); bobo_TAN@scdf.gov.sg (B.T.); Shalini_ARULANANDAM@scdf.gov.sg (S.A.); 2Engineering Product Development Pillar, Singapore University of Technology and Design, Singapore 487372, Singapore; amelia_yeo@mymail.sutd.edu.sg; 3Department of Emergency Medicine, Singapore General Hospital, Singapore 169608, Singapore; marcus.ong.e.h@singhealth.com.sg (M.E.H.O.); sophronesis@gmail.com (A.F.W.H.); 4SingHealth Emergency Medicine Academic Clinical Programme, Duke-National University of Singapore Medical School, Singapore 169857, Singapore; 5Department of Community Emergency Health and Paramedic Practice, Monash University, Clayton, VIC 3800, Australia; karen.smith@ambulance.vic.gov.au; 6MOH Holdings Pte Ltd., Singapore 099253, Singapore; normanhylin@gmail.com

**Keywords:** cardiac arrest, cardiopulmonary resuscitation, residential, urban, high-rise

## Abstract

Out-of-hospital cardiac arrests (OHCAs) occurring in high-rise buildings are a challenge to Emergency Medical Services (EMS). Contemporary EMS guidelines lack specific recommendations for systems and practitioners regarding the approach to these patients. This scoping review aimed to map the body of literature pertaining to OHCAs in high-rise settings in order to clarify concepts and understanding and to identify knowledge gaps. Databases were searched from inception through to 6 May 2021 including OVID Medline, PubMed, Embase, CINAHL, and Scopus. Twenty-three articles were reviewed, comprising 8 manikin trials, 14 observational studies, and 1 mathematical modelling study. High-rise settings commonly have lower availability of bystanders and automatic external defibrillators (AEDs), while height constraints often lead to delays in EMS interventions and suboptimal cardiopulmonary resuscitation (CPR), scene access, and extrication. Four studies found return of spontaneous circulation (ROSC) rates to be significantly poorer, while seven studies found rates of survival-to-hospital discharge (*n* = 3) and neurologically favourable survival (*n* = 4) to be significantly lower in multistorey settings. Mechanical chest compression devices, transfer sheets, and strategic defibrillator placement were suggested as approaches to high-rise OHCA management. A shift to maximising on-scene treatment time, along with bundling novel prehospital interventions, could ameliorate some of these difficulties and improve clinical outcomes for patients.

## 1. Introduction

Emergencies occurring in high-rise buildings are becoming increasingly prevalent due to rapid urbanisation globally and they present significant challenges to prehospital emergency care. This is especially true for out-of-hospital cardiac arrest (OHCA), which is the most time-sensitive medical emergency. OHCA has generally poor survival rates [1], but favourable clinical outcomes are possible if essential care processes are rendered in a rapid and seamless manner, as exemplified by the “chain of survival” model [2]. Previous literature has consistently shown that prehospital interventions confer greater survival impact in OHCA relative to advanced hospital-based interventions, and the benefits of the latter are confined to those who receive timely prehospital interventions [3]. Thus, optimising the efficiency and effectiveness of emergency medical services (EMS) in providing resuscitative care, along with the public response to OHCA occurring in high-rise buildings, is of paramount clinical and scientific interest.

Real-world data from several regions have shown poorer clinical outcomes among OHCAs occurring in high-rise locations [4,5,6]. A study from Singapore went on to demonstrate a dose–response effect in the highly urbanised Southeast Asian city, with survival being lower with incremental floors above the ground [7]. The reasons for this effect are unclear, but the findings of delayed access to patients, increased transport times, and reduced rate of bystander cardiopulmonary resuscitation (CPR) shown in several studies suggest that disruption in the chain of survival (particularly early CPR) is part of the causal pathway [4,5,6]. In densely populated areas where large proportions of the population reside in high-rise residential buildings, EMS crews frequently encounter scene access and stretcher transport difficulties due to narrow corridors and enclosed elevators [5,8]. Rapid urbanisation and densification, which are happening at an increasing pace [9,10], further complicate this issue of vertical access and care delivery for EMS systems.

However, contemporary guidelines for resuscitation and EMS protocols lack specific recommendations for EMS systems and practitioners regarding their approach to patients in cardiac arrest in high-rise buildings. Furthermore, definitions and standards on the classification of high-rise buildings used in the literature are heterogenous, ranging from 3-storey apartment buildings without elevators [11] to those with more than 30 storeys and with elevator access to every floor [12]. 

This scoping review therefore aimed to map the body of literature pertaining to OHCAs occurring in high-rise settings in order to clarify concepts and current understanding, as well as to identify knowledge gaps. The themes investigated were the extent of the problem, outcomes and prognosis, unique challenges, and potential solutions. 

## 2. Materials and Methods

This scoping review protocol was guided by recommendations from Arksey and O’Malley’s framework and the Preferred Reporting Items for Systematic reviews and Meta-Analyses extension for Scoping Reviews (PRISMA-ScR) [13,14] As the study designs and definition of high-rise differed across contexts with no clear indication of homogeneity in the literature, a scoping review, instead of a systematic review, was chosen to give an overarching perspective of the challenges, prognoses, unique approaches, and solutions in caring for OHCA occurring in high-rise settings.

### 2.1. Search Strategy 

In consultation with a medical information specialist, a search strategy was developed employing various combinations of the keywords ((out-of-hospital cardiac arrest OR out of hospital cardiac arrest OR OHCA) AND (high-rise OR high rise OR height* OR vertical* OR skyscraper* OR tall OR elevator* OR stair*)). Five bibliographical databases were searched from database inception through to 6 May 2021: OVID Medline, PubMed, Embase, Cumulative Index to Nursing and Allied Health Literature (CINAHL), and Scopus. Abstracts were screened using Covidence (Melbourne, Victoria, Australia) by three independent researchers (M.X.H., A.F.W.H. and Q.X.N.). Full texts were obtained for all articles of interest and their reference lists were manually searched to identify additional relevant papers. Subject content experts were consulted to identify additional relevant articles. Conflicts were resolved by discussion and consensus amongst the study team (M.X.H., A.F.W.H. and Q.X.N.).

### 2.2. Selection Criteria

Articles were considered eligible for inclusion if they reported on OHCAs in a high-rise building (in order to encompass all relevant papers despite heterogeneity of definitions, we adopted an inclusive definition of any building with individual floors located above ground). All study designs (case reports, case series, randomised controlled trials, and observational cohort studies) were included in the initial search. Subsequently, studies were excluded if they did not present primary data, did not have an accompanying English translation, or had no specific description of the type of high-rise components (i.e., floor levels, staircases, elevators) within the location of arrest. Abstracts with reported data but no full text available were referenced accordingly and their corresponding documents used as the full text. 

### 2.3. Data Extraction 

Relevant quantitative and qualitative data were extracted by two authors (M.X.H. and Y.A.N.) and cross-checked by a third author (A.F.W.H. or Q.X.N.). Categorical variables were presented as percentages while continuous variables were presented as mean and standard deviation (SD), or median and interquartile range (IQR). The data included several outcomes of interest, namely survival to discharge, neurologically intact survival at discharge, return of spontaneous circulation (ROSC), CPR quality measures (compression rate and depth), and operational time intervals between the EMS crew’s arrival to and departure from the scene. A favourable neurological outcome was defined as a cerebral performance category (CPC) score of 1 or 2. 

### 2.4. Ethical Considerations

Ethical approval was not required as this was a scoping review study and did not include any human subjects or participants. 

## 3. Results

Figure 1 shows the study selection process. The database search yielded 183 records, with 4 additional records obtained from secondary sources. A total of 46 studies were removed as duplicates and a further 117 were excluded after title and abstract screening. A further seven articles were removed after review of full texts. Finally, 23 articles were included in the scoping review [4,5,6,7,8,11,12,15,16,17,18,19,20,21,22,23,24,25,26,27,28,29,30]. The characteristics of included studies are summarised in Table 1.

### 3.1. Geographical Distribution of Studies

As shown in Figure 2, studies reporting on high-rise OHCA were mostly found in densely populated and metropolitan regions such as South Korea (*n* = 7), Taiwan (*n* = 3), Singapore (*n* = 2), and Japan (*n* = 1). Apart from these 13 Asian studies, the remaining studies originated from Europe (*n* = 4) and North America (*n* = 6).

### 3.2. Unique Challenges of High-Rise Settings

High-rise OHCA poses a myriad of challenges to EMS personnel and first responders alike due to the vertical height. Most evident are the obstacles to scene access and egress. Travelling vertically within a building involves an additional layer of transport via an elevator or staircase. Certain floors may not have elevator access and occasionally the elevator may be too small for the stretcher [24]. Multiple elevator stops in a building with high human traffic interfere with EMS crew response as EMS responders are generally unable to override the elevator’s mechanism to bypass floors and provide them with the necessary priority [6]. These barriers result in both time delays [18,19,20,26] and difficulty in maintaining CPR quality during transfer to the ambulance [11,21].

OHCAs occurring in high-rise buildings also have a lower chance of being witnessed by a bystander who is able and willing to perform basic life support, as highlighted by Lee et al. (2018), who reported a common lack of trained bystanders in high-rise settings [27]. Compounding this problem is the limited access to defibrillators, which are most often located on the ground floor of high-rise buildings [31]. The additional time needed to fetch the equipment and reach the patient via the elevator is also proportional to the number of elevator stops encountered along the way [6].

A total of seven studies reported an adverse impact of vertically higher locations of arrest on EMS time intervals. With respect to the time between EMS arrival on-scene and arrival at patient’s side, also known as T4 and T5, Silverman et al., Park et al., Morrison et al., Lateef et al., and Choi et al. uniformly reported delays for high-rise OHCA cases [4,5,8,18,26]. Furthermore, the studies by Lateef and Park found that this finding remained applicable to the time interval between leaving the patient’s location and the commencement of the ambulance’s journey to the hospital, known as T6 and T7, respectively [8,26]. In Heidet et al.’s Parisian study, it was found that the number of floors in a patient’s residence significantly affected EMS response times [22]. This was particularly prevalent in the most deprived areas of the precinct with more multistorey dwellings.

Conway et al. was the only study in this review that measured the time interval between arrival at-scene and prehospital defibrillation, termed curb-to-defib interval [19]. It was reported that tall buildings and buildings with larger volumes had significantly greater curb-to-defib intervals as compared to shorter buildings and those with smaller volumes.

There were seven studies, all manikin trials, which suggested that higher floors compromised CPR quality. Bekgoz et al. and Chen et al. reported poorer CPR quality (measured in terms of lower chest compression fractions) with manual compared to mechanical chest compressions when manual compressions were administered to the manikins during transport from the third to first floor [11,15]. Drinhaus et al. similarly reported a significantly lower proportion of good-quality compressions with adequate rate when manual compressions were performed en-route via a lift, turntable ladder, or staircase [21]. When a standard stretcher was used in Kim et al.’s 2016 study, there was a significantly smaller proportion of compression with adequate depth and rate when the manikins were transported from the sixth to first floor [24]. Conversely, Chi et al. found no significant differences in chest compression fraction between manual and mechanical CPR groups when manikins were transported from the thirteenth to first floor [17].

### 3.3. Prognosis and Outcomes

Given the aforementioned challenges, studies have reported congruent findings on the negative impact of high-rise settings on EMS time intervals, CPR quality, and the clinical outcomes of OHCA, namely survival and ROSC.

Three studies reported congruent findings of a negative impact of higher floor number on survival to hospital discharge. Drennan et al. found that patients living three floors and higher above ground had a significantly lower unadjusted survival-to-hospital discharge of 2.6% as compared to those who lived below the third floor (4.2%) [20]. Similarly, Lian et al. reported that the unadjusted rates of survival to hospital discharge declined from 2.7% for patients residing on the second floor to 0.7% for patients on the sixth floor [7]. This difference remained significant after adjustment for confounders. Sinden et al. similarly reported lower rates of survival to hospital discharge in OHCA patients when EMS arrival at scene to patient’s side was delayed [29].

Four studies reported similar findings of the negative impact of higher floors on neurologically intact survival measured at hospital discharge or at 1 month. Kobayashi et al. and Sohn et al. both reported significantly lower unadjusted rates of neurologically intact survival for OHCA patients living 3 floors or higher above ground [25,30]. Choi et al. found that patients residing on a high floor of 3 storeys or more had significantly lower unadjusted rates of neurologically intact survival compared to if the OHCA took place in a public area [18]. Interestingly, if the arrest occurred at home, favourable neurological outcomes were more likely in patients residing on higher floors. Sinden et al. similarly reported lower unadjusted rates of neurologically intact survival for patients subjected to longer EMS arrival times [29].

Five studies reported on the outcomes of prehospital ROSC in their results. Chi et al., Kobayashi et al., and Sohn et al. showed a consistent detrimental effect of higher floors on patient ROSC [16,25,30]. These three studies reported a significantly lower rate of prehospital ROSC for patients living on the third floor and higher, compared to lower floors. In particular, the study by Chi et al. reported an odds ratio of 0.40 (95% CI 0.17–0.98) for ROSC in a vertical OHCA location. Heidet et al. reported that ROSC rates were significantly poorer only for the most deprived areas in a densely populated Parisian precinct, with 33% of buildings being multistorey residential blocks [22]. Contrarily, Choi et al. reported anomalous findings of a significantly higher rate of ROSC for residential OHCAs occurring at higher floors of three storeys and above as compared to lower floors [18]. The study was located in South Korea and defined residential areas as apartments, condominiums, and townhouses. All analyses were unadjusted for potential confounders.

### 3.4. Approaches and Solutions

Given the poorer outcomes of OHCA patients from high-rise settings and poorer quality of prehospital interventions, some studies have attempted to look at solutions to address these issues. Six manikin trials compared the use of mechanical CPR (mCPR) with manual compressions during scenario-based resuscitative procedures in high-rise settings [11,12,15,21,23,28]. Four of these trials reported positive findings for mCPR where its use led to higher chest compression fractions and greater proportions of guideline-compliant chest compression rate and depth [11,12,15,21].

Jorgens et al. was the only trial that compared four different mCPR devices through a multistage route and found that the need for correction of pressure points was the lowest with the use of LUCAS-2 [23]. Finally, only one manikin trial reported the use of an active compression decompression (ACD) device and compared this with load-distributing band mCPR in an elevator setting [28]. It was reported that the use of LUCAS-2 mCPR compressions with the ACD had the lowest percentage change in compression depth and was recommended in elevator settings, which necessitate changes of stretcher positioning.

In terms of introducing specific equipment for procedural transport, two studies reported logistical interventions that improved CPR quality. Kim et al.’s 2016 study employed the use of a reducible stretcher that accommodated a hinged position during transport. CPR in the reducible stretcher group was found to have a significantly higher proportion of good compressions with adequate depth and rate as compared to the standard stretcher group [24].

Alternatively, Chi et al. employed the use of a transfer sheet in their 2020 study. Instead of placing the manikin on a stretcher before entering the elevator, the manikin was lifted with a transfer sheet and placed directly onto the elevator floor. It was reported that this led to significantly better compressions in terms of adequate depth and rate, and significantly shorter time intervals between moving the patient from the scene into the elevator [17].

Moreover, the strategic placement of defibrillators can contribute to a reduction in time to first defibrillation, as reported in Lee et al.’s 2018 study were AED placements in high-rise buildings increased willingness of inhabitants to perform CPR and utilise a defibrillator [27].

Of particular interest is Chan’s 2017 study that developed a mathematical model of a high-rise building equipped with floors, one elevator, and one AED. Based on theoretical calculations, placing AEDs in elevators would benefit buildings only if they were sufficiently tall. If the OHCA risk on the ground floor were higher, such as in buildings with busier street level traffic or underground walkways, a lobby-based AED would be more beneficial [6].

## 4. Discussion

Across the included studies, it is apparent that high-rise settings pose significant challenges to EMS response to OHCA cases, resulting in poorer clinical outcomes for patients. This, however, does not change the fact that high-rise buildings are commonplace in many urbanised cities. Prehospital EMS systems could consider addressing the following gaps in the delivery of care for high-rise OHCA patients.

Firstly, protocols to override elevator systems in emergency situations can mitigate delays to scene access. In 2012, a patent was issued on a method of operating elevators during emergency situations [32]. This involved elevator cars being recalled to the ground floor and temporarily taken out of service till the arrival of emergency medical personnel, who can use a unique key and travel to designated floors within the building to attend to casualties or evacuate residents. Delays to extrication can likewise be reduced with specific equipment such as transfer sheets and stretchers which can accommodate tight spaces, as reported in the manikin trials [17,24], although the actual deployment of these equipment for real-life situations remains to be elucidated.

Secondly, there is value for EMS crews to maximise treatment opportunities, especially during times where patient movement is minimal and where procedural transfers are not yet necessary. The time-critical urgency of OHCAs coupled with the constant reminders for EMS crew to rapidly transport patients to secondary or tertiary care facilities could at times be an albatross that distracts EMS personnel from providing quality and vital ALS treatment on-scene.

Lengthening on-scene time allows EMS services operating in highly urbanised environments to implement novel, bundled interventions and team-led high-performance CPR. Ambulance services in well-developed EMS systems, such as that of Victoria, Australia, have formalised a delay in mechanical CPR during the crucial early stages of resuscitation. This underlines the importance of staying on-scene for the delivery of optimal basic and advanced life support to achieve ROSC [33]. Liao et al.’s 2019 manikin trial reported the use of an ACD together with a LUCAS-2 mechanical CPR device, which led to better CPR quality in the elevator. This bundling of interventions has been found to significantly improve cerebral perfusion pressure in porcine studies of cardiac arrest [34], while other interventions such as head-up CPR and an impedance threshold device (ITD) have also been reported as effective solutions as part of an optimal bundle of OHCA management [35,36,37]. While these have yielded promising findings in porcine models, the transferability of such interventions to real-world OHCA patients in a dynamic prehospital environment remains to be elucidated.

Thirdly, timely CPR may be better achieved with improved public education programmes, especially for family members of at-risk patients (e.g., chronic heart failure). Given that bystander and lay rescuer involvement have been highlighted as a potential issue in high-rise settings, systemic, nationwide strategies that leverage technology to bring trained rescuers closer to OHCA victims are promising steps forward in tackling the challenge of high-rise OHCAs. In Singapore, the myResponder application is used to notify trained responders of cardiac arrest cases within a 400 m radius and the location of the nearest defibrillator. The responder may be located on a different floor in the same high-rise building, but would still be able to respond swiftly [38]. This concept of training members of the public and utilising them as prehospital manpower is a promising approach, and is practised in a similar fashion in London with the GoodSAM application [39] and also echoed by the European Resuscitation Council in their statement on teaching CPR to children in schools [40].

The prudent deployment of trained first responders in high-rise buildings in the form of fire or cardiac arrest wardens could also augment early access to OHCA victims. Defibrillators could also be issued to first responders who are constantly on the move, such as train drivers or drivers of hired cars and taxis. A recent collaboration between the Singapore Civil Defence Force and the Singaporean private car hire company Grab equipped private hire drivers with AEDs as part of the AED-on-Wheels programme [41]. This allows drivers to respond swiftly to any location when notified of an incident within their radius.

Fourthly, the concept of energy ratings for buildings can perhaps be extended to health ratings in the context of prehospital measures such as presence of AEDs, cardiac arrest wardens, and trained responders as a percentage of the resident population. This evaluation of a building’s safety can be applied to residential and nonresidential spaces and shed light on the areas that need more robust AED deployment and bystander training.

Lastly, given that contemporary EMS response time data have been mainly centred on call-to-curb intervals which lack data on building height, floor levels, and elevator or AED availability, there could be value in the creation of prehospital datasets that are unique to high-rise settings. Variables such as the number of storeys and even the type of residential unit could be recorded as part of ambulance case records and transferred to registry data. This information could furnish valuable insight into space constraints and the difficulties impeding the smooth execution of team resuscitative procedures due to certain factors such as the lack of a 360 degree access and overview of the patient.

As an extension to unique prehospital OHCA datasets, a linkage of OHCA data registries with other data repositories such as socioeconomic data or urban density data could prove useful in the analysis and evaluation of OHCA resuscitative performance. Heidet et al.’s 2020 study successfully retrieved census data on socioeconomic status stratified by geographical areas, and linked that with a validated calculation of the degree of deprivation associated with each of these areas [22]. If such linkage of data is replicated in more EMS systems worldwide, the findings could provide important insight to the unique but less-reported-on barriers EMS systems face in different geographical contexts and demographic groups, hence informing policy changes and improvements.

### Limitations

The findings of this scoping review are limited by retrospective observational evidence and manikin-dominated trial designs. Compared to manikins, compression depth and rate inevitably differs in accuracy when measured on human patients of varying body weight and height. Secondly, while the variable of EMS time intervals has been quantified and analysed in a number of studies, other reported barriers of high-rise settings have been primarily studied in a qualitative manner. Further randomised, controlled trials (RCTs) with human subjects should be conducted to ascertain the efficacy of proposed strategies as well as the impact of high-rise buildings on OHCA clinical outcomes.

## 5. Conclusions

High-rise OHCAs are a challenge for prehospital EMS crew and care systems due to often ineluctable delays in scene access and egress and inherent space constraints. A focus on maximising on-scene treatment time, along with bundling novel prehospital interventions, could ameliorate some of these difficulties and improve patient outcomes.

## Figures and Tables

**Figure 1 jcm-10-04684-f001:**
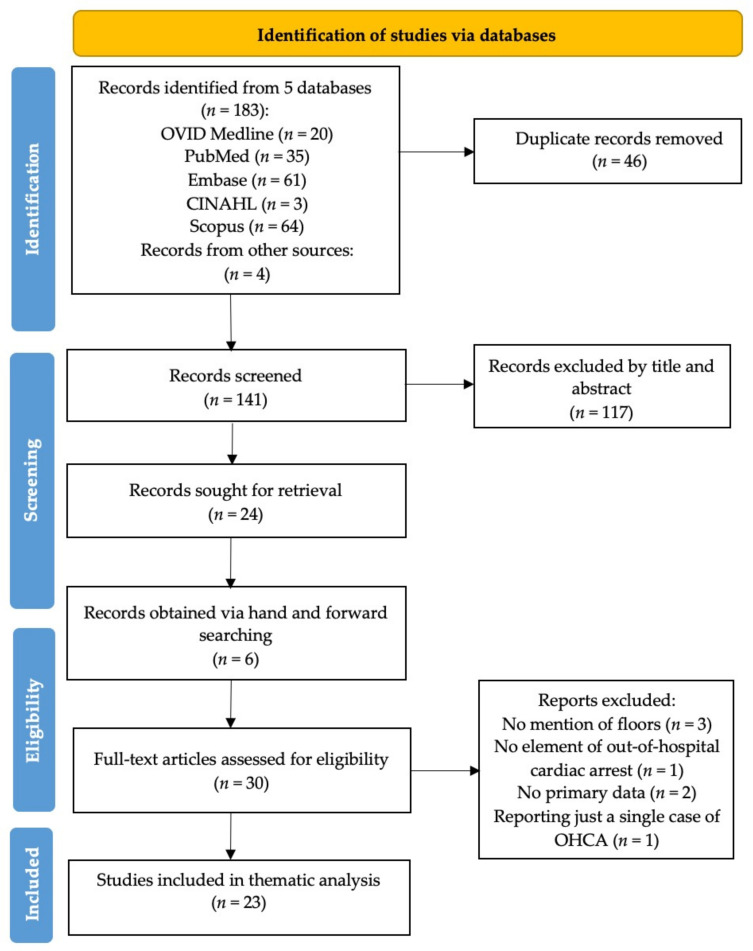
PRISMA flowchart illustrating the study selection process.

**Figure 2 jcm-10-04684-f002:**
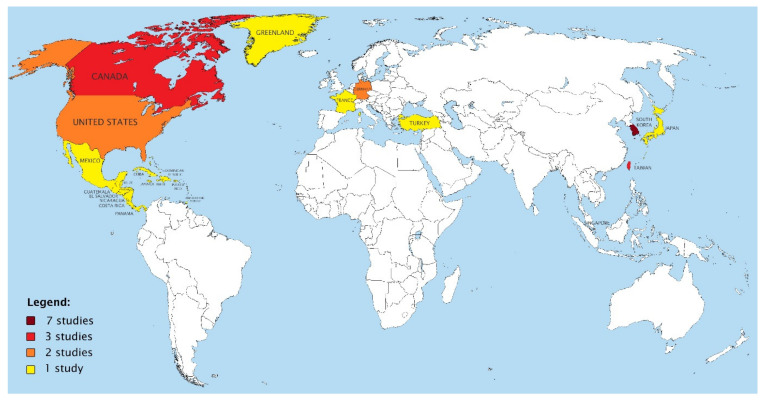
Geographical distribution of studies included in this review.

**Table 1 jcm-10-04684-t001:** Characteristics of included studies.

Author, Year	Country and Study Setting	Study Design (Sample Size)	Study Outcomes	Interventions and Control Groups for Comparison	Results	Conclusions
Bekgoz et al., 2020 [11]	Ankara, capital of Turkey Training centre # floors = 3 Starting floor: 3rd Ending floor: 1st	Manikin trial (*n* = 10 female and 10 male paramedics)	Compression rate (compressions/min) Compression depth (mm) Hands-on time (%)	Manual CPR and manual chest compression device (MCCD)	Median chest compression rate: Higher for manual CPR at 142 compressions/min than the MCCD at 102.3 compressions/min (*p* < 0.01) Median chest compression depth: More shallow for manual CPR at 25.2 mm than MCCD at 52.0 mm (*p* < 0.001) Hands-on time: 92.0% for manual CPR vs. 100% for MCCD (*p* = 0.09)	While transporting the patient simulator manikin to the lower floor, it was found that MCCD achieved high-quality CPR targets recommended by resuscitation guidelines in terms of compression rate, depth, and hands-on time.
Chan, 2017 [6]	Toronto, Canada	Mathematical model of a high-rise building (n floors, single elevator and single AED) OHCA occurrences modelled using independent Poisson processes on each floor	Average override-to-arrival response distance, E(D_OA_) Maximum response distance, max(D_OA_)	Elevator-based AED vs. lobby-based AED Arrest floor I = 0 vs. arrest floor I ≠ 0.	Average response distance was shorter for elevator-based AED if the number of floors exceeded ¾ of the ratio of ground-floor OHCA risk to above-ground floor risk plus 0.5. If not, a lobby-based AED had a shorter response distance. If the risk of OHCA was equal for each floor, an elevator-based AED would have a shorter average response distance.	Cardiac arrests in a tall building may experience faster response from an elevator-based AED, whereas a building with much higher risk on the ground floor compared to any above-ground floor would be better off with a lobby-based AED.
Chen et al., 2021 [15]	Taoyuan, Taiwan Environmental conditions: 5-storey building without an elevator. Start: 3rd floor End: 1st/ground floor	Nonrandomised manikin simulation trial (*n* = 20 EMTs placed in 10 pairs) 2 simulation runs per experimental arm with Resusci Anne First Aid full body manikin (60 kg)	CPR quality	Experimental group: mechanical compressions with adapted LUCAS-2 device strapped to manikin before transport down the stairs on a stair stretcher Control group: manual chest compressions with manikin strapped directly to stair stretcher	There were no statistically significant differences in CPR quality between experimental and control groups for the overall resuscitation period. Chest compression fraction: Statistically significantly higher in LUCAS-2 experimental group at 0.76 (0.75, 0.78) vs. 0.63 (0.62, 0.66) for the control group	LUCAS-2 mechanical CPR provides better chest compression fractions than manual CPR in stairwells.
Chi et al., 2016 [16]	Gangdong-gu, Seoul, South Korea	Retrospective observational study (*n* = 119 rescue records) Qualitative survey (*n* = 54 paramedics)	ROSC rate Survival to hospital admission	Vertical location of arrest Manual CPR vs. mechanical CPR	ROSC: Significantly lower likelihood of ROSC in a vertical location of cardiac arrest (OR: 0.40, 95% CI 0.17–0.98, *p* = 0.044) Survival to hospital admission: No statistically significant differences between OHCA patients from ground floors vs. nonground floors Qualitative survey: 85.2% felt there was a lower quality of chest compressions for patients above ground floor, and 93.1% felt that mechanical CPR devices could circumvent this	Vertical location of cardiac arrest scene independently affects ROSC rates in OHCA.
Chi et al., 2020 [17]	Taipei, Taiwan Conventional passenger elevator: Length = 1.6 m Width = 1.5 m Height = 2.2 m Start: 13th floor End: 1st/ground floor	Randomised open-label cross-over manikin trial (*n* = 72 simulation runs with EMTs in 12 3-person crews)	Primary outcomes: mean compression depth, chest compression fraction (CCF) Secondary outcomes: Percentage of fully released compressions, compressions with adequate rate, hand position	Before entering elevator: Chest compressions and defibrillation on scene Intervention group: Move manikin to transfer sheet (TS) and enter elevator Stretcher groups: Move manikin to stretcher and adjust to either 45 or 90 degree incline	Mean compression depth: Significantly deeper for TS (54.4 ± 4.2) compared to stretcher groups at 45 degrees (39.6 ± 7.2) and at 90 degrees (40.6 ± 8.3) Chest compression fraction: No significant differences between intervention and control groups Compressions with adequate recoil: No significant differences between intervention and control groups Compressions with adequate depth: Significantly higher for TS at 51% (23.8–74.8) compared to stretcher groups at 45 degrees (19.5, 5.8–29.5) and 90 degrees (25.5, 20.0–34.3) Time interval for moving patient from scene to elevator: Significantly shorter for TS at 22.5 s (18.7–26.8) compared to stretcher groups at 45 degrees (45.3 s, 39.9–52.1) and 90 degrees (47.7 s, 42.1–65.0)	TS is better than stretcher with regards to chest compression depth and transport time. TS should be used for high-rise building transport of OHCA patients. Rescuers need to be aware to allow full chest recoil.
Choi et al., 2019 [18]	Korea, 18 urban and suburban areas Home = apartment, condominium, house, townhouse. Public place = everything else High floor ≥3rd Low floor = 2nd and lower	Retrospective cohort study (*n* = 6355 OHCA cases)	Primary outcome: Neurologically favourable outcome after a high-floor OHCA, measured at hospital discharge (CPC 1 or 2) Secondary outcomes: ROSC, call-to-scene time and call-to-patient time	High-floor vs. low-floor groups	Neurologically favourable discharge: Significantly lower for high-floor OHCA if the incident occurred in a public place (aOR, 0.58; 95% CI: 0.37–0.89) Significantly higher for high floor OHCA if incident occurred at home (aOR, 1.40; 95% CI: 0.96–2.03) ROSC:Significantly higher for home OHCAs occurring at high floors (551, 25.39%) compared to lower floors (421, 21.22%), *p* = 0.002 Time intervals: The call-to-scene time was a median of 7 min, which was shorter on a high floor for events in both homes and public places. Call-to-patient time for home events was significantly longer on a high floor (a median of 9 min).	Nature of setting (home vs. public) affects EMS response times to OHCA in high-rise buildings. Patient’s prognosis is more likely to be affected by the structure and use of the building, rather than the floor height where the CA event occurred.
Conway et al., 2016 [19]	Seattle, Washington, United States. 2-tiered EMS response system with BLS fire engine and ALS ambulance Buildings were categorised as short (<25ft), medium (26–64 ft), and tall (>64 ft)Building volumes were categorised as small (<60,000 ft^3^), midsize (60,000–1,202,600 ft^3^) and large (>1,202,600 ft^3^)	Retrospective observational study (*n* = 3065 OHCA cases that occurred indoors and without prior deployment of defibrillator)	Call-to-curb interval: call to on-scene time Curb-to-defibrillation interval: on-scene to defibrillation on time Call-to-defibrillation interval	Time-intervals against BHC Time-intervals against BVC	Median call-to-curb intervals by building height and volume: Significantly lower for tall buildings (3.96) compared to short and medium-height buildings (4.73, 4.27), *p* < 0.01 Significantly lower for larger-volume buildings (4.05) compared to smaller-volume buildings (4.87), *p* < 0.01 Median curb-to-defib intervals by building height and volume: Significantly greater for tall buildings (3.11) compared to short and medium-height buildings (1.97, 2.62), *p* < 0.01 Significantly greater for larger-volume buildings (3.01) compared to smaller- and medium-volume buildings (1.90, 2.58), *p* < 0.01 Median call-to-defib intervals by building height and volume: No significant differences between any groups	Buildings with greater height and volume have significantly longer curb-to-defib times and significantly shorter call-to-curb times. The hypothesis that taller or larger-volume buildings cause poorer outcomes was not supported by this study’s results.
Drennan et al., 2016 [20]	Toronto, Canada OHCAs that occurred within Toronto city and the regional municipality of Peel. Floor of patient contact: High ≥ 3rd Low ≤ 2nd Private locations: apartments, condominiums, houses, or townhouses. Others = public/other	Retrospective observational study (*n* = 7842 OHCA cases)	Primary outcome: Survival to hospital discharge Secondary outcomes: Delay to patient contact, use of AEDs by bystanders	Low floors (<3 floors) vs. high floors (3 floors and higher)	Survival to hospital discharge: Significantly lower for patients residing on the third floor and above (2.6%) as compared to below the third floor (4.2%), *p* = 0.002 Time interval between arrival and patient contact:Significantly greater for higher floors compared with lower floors (4.9 ± 2.6 vs. 3.0 ± 2.0, *p* = 0.01) Use of AED: No significant differences although the rate of use was very low regardless of floor level (0.3% for lower floors and 0.4% for higher floors)	OHCA on high floors had lower rates of survival to hospital discharge, and no survivors above the 25th floor. This is likely due to longer intervals from arrival of 911 responder to patient contact, and lower rates of initial shockable rhythm for high floor patients.
Drinhaus et al., 2020 [21]	Brühl, North Rhine-Westphalia, Germany.Test setting: apartment (5th floor), evacuated to ground floor via lift, turntable ladder, or staircase	Manikin trial (*n* = 40 paramedics	CPR quality measures:Compression depth and frequency	6 experimental arms: Lift and manual CPR Lift and mechanical CPR Ladder and manual CPR Ladder and mechanical CPR Staircase and manual CPR Staircase and mechanical CPR	Percentage of guideline-compliant CPR compression depth and frequency for lift route: No significant differences for depth. For frequency, significantly lower manual CPR compared to mechanical CPR (58 ± 34 vs. 94 ± 2, *p* = 0.02) Percentage of guideline-compliant CPR compression depth and frequency for ladder route: Significantly lower for manual CPR compared to mechanical CPR (depth: 18 ± 21 vs. 92 ± 7, *p* = 0.04; frequency: 61 ± 17 vs. 96 ± 1, *p* = 0.04) Percentage of guideline-compliant CPR compression depth and frequency for staircase route: Significantly lower for manual CPR compared to mechanical CPR (depth: 25 ± 16 vs. 86 ± 28, *p* = 0.02; frequency: 22 ± 30 vs. 96 ± 2, *p* = 0.02)	Mechanical CPR is more effective in delivering consistent high-quality CPR regardless of floor level. Manual CPR results in inconsistent CPR quality particularly for ladder and stairs and may pose possible hazards to staff.
Heidet et al., 2020 [22]	Val-de-Marne, Paris, France Population (2012) of 1,365,039 inhabitants, mean population density of 5572 inhabitants per square kilometer SAMU (Service d’aide médicale urgente) dispatches 6 hospital-based physician-staffed EMS ambulances (Service mobile d’urgence et reanimation or SMUR)	Multicentre prospective cohort study *n* = 2298 cases of SMUR dispatch	Primary Outcome: Overall EMS response time interval (time from vehicle start to patient contact) Secondary Outcomes: Vehicle time interval Patient access time interval ROSC Survival on scene Survival to 30 days	Area-level socioeconomic status assessed at census level. SES evaluated using French version of European Deprivation Index (French-EDI) Quintile 1: least deprived Quintile 5: most deprived	There were more floors in the most deprived areas, along with more frequent access barriers and younger age of patients. EMS response times were all significantly affected by reason for dispatch, dispatch time and location, number of floors in a patient’s dwelling, among other related barriers. Dispatch time, location, number of floors, and post-ambulance stop barriers were associated with patient access time interval. Significant associations between poor OHCA outcomes were only found for the most deprived areas (Quintile 5) In Quintile 5, 28.5% had 1–3 floors while 21.7% had 4 and more floors. 33.1% were multistorey residential buildings.	The more deprived an area was, the longer EMS response times were due to the higher prevalence of EMS access constraints.
Jorgens et al., 2021 [23]	Munchen, Germany	Simulator-based randomised trial 9 paramedics 4 emergency physicians 4-person transport teams to carry mannequin with mCPR through a predefined 10-step transport route	Stability of device (displacement measurement, correct pressure point) Compliance to ERC cardiac massage guidelines (50–60 mm compression depth, 30:2 compression:ventilation ratio, 100–120 bpm) Questionnaire to evaluate each mCPR device using VAS (0 = totally unsuitable, 10 = ideally suited) Questionnaire to rank perceived physical effort using modified BORG CR-10 scale (0 = no exertion/breathlessness, 10 = maximum exertion/breathlessness forces stop)	Control Route: Stationary mCPR at the beginning and end of transport Experimental Route: Transport with soft stretcher and staircase Vehicular trips with turntable ladder, rescue basket Ambulance Transport Loading and Unloading 4 different mCPR devices:animax mono, AutoPulse, corpuls CPR, LUCAS 2	Correct pressure point recorded for 2 out of 15,962 compressions (0.013%) during the application of corpuls CPR in soft stretcher transport Compliance to compression depth had greater scattering during basic resuscitation across all mCPR devices Use of all mCPR devices showed a high level of user satisfaction regardless of route category	It is crucial to check stability of mCPR devices regularly and correct connection to the patient under different transport circumstances.
Kim et al., 2016 [24]	Seoul, Korea Simulation scenarios made up of 3-personnel teams of EMTS and resuscitation manikin placed on the 6th floor of a multistorey building using a small elevator	Randomised crossover manikin simulation trial 9 EMTs in teams of three 44 simulation scenarios with at least 21 trials for each protocol	Primary outcome: No-flow fraction during simulation scenario Secondary outcome: Proportion of chest compressions with adequate compression depth Proportion of chest compressions with adequate compression rate	Preparation: Manual CPR for 3 cycles of 2 min each Standard stretcher group (SS-CPR): Movement of manikin to standard stretcher ETI with manual CPR Standard stretcher transformed to wheelchair position during elevator transport Reducible stretcher group (RS-CPR): Movement of manikin to reducible stretcher and mounting of mCPR ETI with mCPR Reducible stretcher hinged during elevator transport to accommodate head-up, leg-raised position	Median no-flow fraction (%): Higher in SS-CPR vs. RS-CPR (32.9 (31.1–34.9) vs. 31.6 (30.8–32.7), *p* = 0.14) Proportion of chest compressions with adequate depth (%): Significantly higher in RS-CPR vs. SS-CPR (97.8 (88.6–98.6) vs. 83.7 (81.3–84.6), *p* < 0.01) Proportion of chest compression with adequate rate (%): Significantly higher in RS-CPR vs. SS-CPR (95.9 (90.6–98.9) vs. 92.9 (86.1–93.9), *p* = 0.05)	There was significantly higher CPR quality when a reducible stretcher was used with mCPR during vertical transport in a high-rise building, specifically for nonflow fraction and proportion of adequate chest compression depth and rate.
Kim et al., 2018 [12]	Jeonju, South Korea Simulation environment was a residential unit located in a 42-storey (180 m) high-rise building. Elevator had a 15-person capacity and travelled at 180 m/min. Door width was 900 mm and internal area was 1600 × 1500 mm. Flexible stretcher (DA-02768 Delti medical, Taiwan) was used for transport within the building	Randomised manikin simulation trial *n* = 24 professional EMS providers 4 scenarios Phase 1: initiating until performing CPR according to 1 of 4 scenarios Phase 2: leaving the scene to entering elevator Phase 3: entering till exiting elevator Phase 4: exiting elevator till loading into ambulance	Average compression depth and rate Incomplete chest recoil ratio Total flow time fraction and flow time fraction by phase of trial Duration of compression pauses	Initial preparation: 5 cycles of 30 compressions and 2 ventilations with defibrillation MAB scenario: 30 manual compressions and 2 BVM ventilations MAS scenario: 30 manual compressions and 2 SGA ventilations MED scenario: continuous mCPR compressions and BVM ventilation MES scenario: continuous mCPR compressions and SGA ventilation Post-scenario:transport to elevator on flexible stretcher, loading into ambulance	No-flow time due to artificial ventilation: Significantly shorter in SGA groups (MAS 49.6 ± 6.0 and MES 52.8 ± 7.9) as compared to BVM groups (MAB 66.2 ± 12.6 and MEB 73.6 ± 9.2) Interruptions due to manikin movement: Significantly shorter in mechanical groups (MEB 3.4±4.3 and MES 3.3 ± 7.9) as compared to manual groups (MAB 76.8 ± 14.7 and MAS 70.2 ± 10.9) Flow time: Highest in MES groups through phases 2, 3 and 4 Flow time fraction by phase: Highest in MES throughout phases 1 to 4	The quality of manual compressions can be maintained when providing CPR for cardiac arrest patients in a high-rise setting. The use of both a mechanical CPR device and a supraglottic airway increased flow time the most effectively.
Kobayashi et al., 2016 [25]	Osaka, Japan	Prospective cohort study (*n* = 2979 OHCA patients) High-floor group: *n* = 1094 Low-floor group: *n* = 1885	Primary outcome: 1-month survival with neurologically favourable outcome (CPC 1 or 2) Secondary outcomes: Prehospital ROSCAdmission to hospital Survival to one month	High-floor group: OHCA patients residing on 3 or more floors Low-floor group: OHCA patients residing on fewer than 3 floors	Neurologically intact survival at 1-month (CPC 1 or 2): Significantly lower for high-floor group than low-floor group (30 (2.7%) vs. 91 (4.8%), *p* = 0.005) Prehospital ROSC: Significantly lower for high-floor group than low-floor group (77 (7.0%) vs. 188 (10.0%), *p* = 0.007) Hospital admission: Significantly lower in high-floor group than low-floor group (218 (19.9%) vs. 457 (24.2%), *p* = 0.007) One-month survival: Significantly lower in high-floor group than low-floor group (54 (4.9%) vs. 138 (7.3%), *p* = 0.011)	Survival at one month with neurologically favourable outcome was significantly lower for OHCA patients in the high-floor group as compared to the lower-floor group.
Lateef et al., 2000 [26]	Singapore A high-rise building was taken to be one that crew had to ascend at least one flight of stairs i.e., 2nd storey and higher. Ground-level building did not involve any stair climbing.	Prospective cohort study (*n* = 150 ambulance runs)	Arrival-to-patient contact delay	Time interval between T4 (time ambulance arrives at scene) and T5 (time at patient’s side) Time interval between T6 (time of leaving location) and T7 (time when ambulance starts journey to hospital)	Mean delay between T4 and T5: Significantly higher in high-rise group as compared to ground-level group. (2.49 ± 0.98 vs. 1.02 ± 1.41, 95% CI: 1.20, 1.75 min, *p* = 0.0106) Mean delay between T6 and T7: Significantly higher in high-rise group as compared to ground-level group. (3.24 ± 1.58 vs. 1.27 ± 0.71, 95% CI: 1.68, 2.04 min, *p* = 0.0098) Questionnaire findings: - Presence of elevator stops only at particular storeys - Multiple stops elevator was put through due to other members of public - Elevator not immediately available for crew - No person to direct crew upon arrival at elevator landing - Narrow elevator landing and stairways	High-rise buildings lead to significant delays to patient access and evacuation to hospital.
Lee et al., 2018 [27]	Daegu, South Korea Definition of high-rise not specified	Cross-sectional survey (*n* = 1445 high-rise apartment managers)	Willingness of apartment managers to perform CPR and use an AED	Apartment managers who worked in apartments with AEDs vs. those who worked in apartments without AEDs	AED placement in high-rise apartments increased willingness to CPR (OR, 1.33; 95% CI: 1.04–1.71) and increased willingness to use an AED (OR, 1.39; 95% CI: 1.10–1.75).	Apartment managers will benefit from AED placement in high-rise buildings and refresher courses on CPR to maintain CPR skills.
Lian et al., 2019 [7]	Singapore	Retrospective cohort study (*n* = 5678 OHCA cases from 01 January 2011 to 31 December 2014)	Primary outcome: 30-day post-cardiac arrest survival or survival to hospital discharge	Floor/floor groups: −4 to −1, 1, 2, 3, 4, 5, 6, 7, 8, 9, 10, 11, 12, 13, 14, 15, 16–20, 21–25, >25	Survival rates: 4.5% for 4 basement floors (−4 to −1) vs. 6.2% for ground floor 2.7% at floor 2, declining to 0.7% at floor 6 Both linear and quadratic floor effects remained significant after adjusting for other confounders (age, bystander witnessed, EMS response time).	There is a significant U-shape relationship between vertical location and OHCA survival, even after adjusting for other OHCA variables. Midrange floors had lower rates of survival to 30 days as compared to basements, ground floor and extreme upper floors.
Liao et al., 2019 [28]	Taipei, Taiwan Elevator setting with standard stretcher	Triple-arm, prospective manikin simulation study (*n* = 12 paramedic teams and 44 simulation runs)	No flow-fraction Time to first shock Percentage change in compression depth between supine and head-up stretcher positions	Conventional CPR (C-CPR) Load-distributing band mCPR (LDB) mCPR with Autopulse Active compression-decompression (ACD) mCPR with LUCAS-2	No-flow fraction: Significantly lower in ACD group (9.6%, 95%CI: 8.5–10.8%) than C-CPR group (28.6%, 95%CI: 25.9–31.4%) and LDB group (14.9%, 95%CI: 13.6–16.2%) Percentage change in compression depth during stretcher position change: Significantly lower in ACD group (2.6%, 95%CI: 1.8–3.3%) than C-CPR (31.2%, 95%CI: 25.7–36.8%) group and LDB group (7.1%, 95%CI: 5.9–8.3%), *p* < 0.001	ACD-CPR is recommended for use in an elevator to improve CPR quality as it was shown to outperform other options in terms of no-flow fraction and percentage change in compression.
Morrison et al., 2005 [5]	Toronto, Ontario, Canada	Observational study (*n* = 118 EMS calls) Single third-party EMT-P observer followed on ambulance runs and recorded data.	Patient access time interval: defined as time of ambulance arrival at scene (vehicle stops) to time of physical contact with patient (at patient’s side) Barriers to paramedic movement (qualitative)	Fewer than 3 floors above or below ground compared with 3 or more floors above ground	Median patient access time interval: Significantly higher for patients located 3 or more floors above ground (2.73 (2.22, 3.03)) as compared to lower levels (1.25 (1.07, 1.55). Significantly greater when patients resided in apartments (2.12 (1.70, 2.50)) as compared with all other buildings (1.02 (0.65, 1.47)), *p* < 0.001. Barriers in apartment buildings (*n* = 40): security code entry requirements (67%), lack of directional signs (82.6%), inability to fit stretcher into elevator (67.9%)	Ambulance calls to places three or more floors above ground had significantly longer patient access time intervals, which make up a substantial part of total EMS response time.
Park et al., 2010 [8]	Seoul, South Korea	Prospective study *n* = 35 ambulance runs *n* = 11 for ground-level group and *n* = 24 for high-rise group	Time interval between T4 (time ambulance arrives at scene) and T5 (time at patient’s side) Time interval between T6 (time of leaving location) and T7 (time when ambulance starts journey to hospital)	High-rise group (more than 1 floor above ground) vs. ground-level group	Median time interval (min) between T4 and T5: Significantly higher for the high-rise group as compared to the ground-level group (2.08 vs. 0.34, *p* = 0.000) Median time interval (min) between T6 and T7: Significantly higher for the high-rise group as compared to the ground-level group (3.08 vs. 1.00, *p* = 0.000) Narrow elevators were cited as the most frequent access barrier (100%) in all of the 24 high-rise ambulance runs	There were significantly longer time delays in the access and evacuation of patients in high-rise buildings.
Silverman et al., 2007 [4]	New York City, United States	Prospective observational case series (*n* = 449 ambulance calls between July 2001 and December 2003)	Time interval from arrival on-scene to patient’s side	Different location settings: multistorey residence, private home (<4 storeys), office building, street, train station, store/mall	Time interval (min) by location: Significantly longer for multistorey residential buildings as compared to private homes or residential areas with 3 or fewer storeys (2.8 vs. 1.3, *p* < 0.0001) Time interval (min) by building height: Significantly longer for office, apartment or medical facilities 10 storeys and higher (3.2 (IQR, 2.7–4.2)) as compared to buildings less than 10 storeys (2.3 (IQR, 1.6–3.1)), *p* < 0.0001.	Patients located in multistorey buildings are subjected to longer vertical response time intervals, which accounts for a big portion of the overall EMS response time in a large metropolitan area.
Sinden et al., 2020 [29]	North America	Secondary analysis of the dataset from the Resuscitation Outcomes Consortium “Trial of Continuous or Interrupted Chest Compressions During CPR” 24,685 case data included in study	Primary outcome: neurologically intact survival at hospital discharge (mRS 3 or less) Secondary outcome: survival to hospital discharge	Curb-to-Care (CTC) interval quartiles (seconds) defined as time interval between EMS vehicle arrival at scene and patient’s side 63–115 116–180 ≥181 compared with ≤62	Neurologically intact survival at hospital discharge: Lower rates of neurologically intact survival for longer CTC quartiles (63–115, 116–180 and ≥181) with adjusted ORs 0.95, (95% CI 0.83–1.09); 0.77 (95% CI 0.66–0.89); 0.66 (95% CI 0.56–0.77) respectively. Survival to hospital discharge: Lower rates of survival to hospital discharge for longer CTC quartiles (63–115, 116–180 and ≥181) with adjusted ORs 0.92, (95% CI 0.81–1.05); 0.79 (95% CI 0.68–0.89); 0.67 (95% CI 0.58–0.78) respectively.	Lower CTC intervals improve patient outcomes in out-of-hospital cardiac arrests. A 2-min CTC interval could be a quality improvement target.
Sohn et al., 2020 [30]	Seoul, South Korea	Retrospective study (*n* = 1541 OHCA patients between 1 Oct 2015 and 30 Jun 2018)	Primary outcome: Neurologically intact survival to hospital discharge (CPC 1 or 2)Secondary outcomes: prehospital ROSC, hospital admission, hospital discharge	First and second floor (*n* = 887) compared with third floor and above (*n* = 654)	Neurologically intact survival at hospital discharge: Significantly lower rate for third floor and above as compared to first and second floors (5.2% vs. 11.1%, *p* < 0.001) Prehospital ROSC: Significantly lower rate for third floor and above as compared to first and second floors (9.9% vs. 16.4%, *p* < 0.001) Median EMS on-scene time (time interval between scene arrival and leaving for hospital): Significantly longer for third floor and above as compared to first and second floors (16 min vs. 12 min, *p* < 0.001)	Patients residing on higher floors have less favourable outcomes in out-of-hospital cardiac arrest.

Abbreviations: cerebral performance category: CPC; cardiopulmonary resuscitation: CPR; emergency medical services: EMS; manual chest compression device: MCCD; mechanical cardiopulmonary resuscitation: mCPR; out-of-hospital cardiac arrest: OHCA; return of spontaneous circulation: ROSC.

## Data Availability

The data that support the findings of this study are available from the corresponding author upon reasonable request.

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
