# Peer review of "Cardiac Arrest Occurring in High-Rise Buildings: A Scoping Review"

_jcm, 2021, doi:10.3390/jcm10204684_

Round 1
Reviewer 1 Report
Overall, this scope review should be revised/improved as described below
- Following recently published PRISMA Extension for Scoping Reviews (PRISMA-ScR): Checklist Ann Intern Med. 2018, there are several items that will help standardized this excellent Scoping Reviews
- Introduction: Please summarize rationale/and add objectives of this review to help readers follow this review easier.
- Before conclusion, if possible, please provide limitations of current evidence/scoping reviews and potential implications and/or next steps. - Quality assessment for each included study should be performed and provided.
- When Pubmed is used for the search, MESH terms are always recommended to be included.
- Search terms in OVID Medline, PubMed, Embase, Cumulative Index to Nursing and Allied Health Literature (CINAHL) and Scopus are different. Please attach syntax used in each database as supplementary.
- Add citations of each study in the table.
- Figure 2 need to enlarge with higher resolution
Author Response
Thank you for the comments.
1. We have improved our introduction section as advised and outlined the study limitations in a subsection under discussion.
2. As per scoping review guidelines (Checklist Ann Intern Med, 2018) risk of bias assessment is not applicable for scoping reviews. Moreover, most of the studies in the present review were mannikin trials, hence we chose to omit this in our scoping review.
3. We have appended our full search strategy in the supplementary materials.
4. OVID Medline:
out of hospital cardiac arrest.mp. [mp=title, abstract, original title, name of substance word, subject heading word, floating sub-heading word, keyword heading word, organism supplementary concept word, protocol supplementary concept word, rare disease supplementary concept word, unique identifier, synonyms]
Out-of-Hospital Cardiac Arrest/
OHCA.mp.
1 or 2 or 3
high-rise.mp. [mp=title, abstract, original title, name of substance word, subject heading word, floating sub-heading word, keyword heading word, organism supplementary concept word, protocol supplementary concept word, rare disease supplementary concept word, unique identifier, synonyms]
high rise.mp. [mp=title, abstract, original title, name of substance word, subject heading word, floating sub-heading word, keyword heading word, organism supplementary concept word, protocol supplementary concept word, rare disease supplementary concept word, unique identifier, synonyms]
(height or vertical).mp. [mp=title, abstract, original title, name of substance word, subject heading word, floating sub-heading word, keyword heading word, organism supplementary concept word, protocol supplementary concept word, rare disease supplementary concept word, unique identifier, synonyms]
skyscraper.mp. [mp=title, abstract, original title, name of substance word, subject heading word, floating sub-heading word, keyword heading word, organism supplementary concept word, protocol supplementary concept word, rare disease supplementary concept word, unique identifier, synonyms]
tall.mp. [mp=title, abstract, original title, name of substance word, subject heading word, floating sub-heading word, keyword heading word, organism supplementary concept word, protocol supplementary concept word, rare disease supplementary concept word, unique identifier, synonyms]
5 or 6 or 7 or 8 or 9
4 and 10
PubMed:
("out of hospital cardiac arrest"[MeSH Terms] OR ("out of hospital"[All Fields] AND "cardiac"[All Fields] AND "arrest"[All Fields]) OR "out of hospital cardiac arrest"[All Fields] OR ("out"[All Fields] AND "hospital"[All Fields] AND "cardiac"[All Fields] AND "arrest"[All Fields]) OR "out of hospital cardiac arrest"[All Fields] OR ("out of hospital cardiac arrest"[MeSH Terms] OR ("out of hospital"[All Fields] AND "cardiac"[All Fields] AND "arrest"[All Fields]) OR "out of hospital cardiac arrest"[All Fields] OR ("out"[All Fields] AND "hospital"[All Fields] AND "cardiac"[All Fields] AND "arrest"[All Fields]) OR "out of hospital cardiac arrest"[All Fields]) OR "OHCA"[All Fields]) AND (("high"[All Fields] AND "rise"[All Fields]) OR "high-rise"[All Fields] OR ("vertical"[All Fields] OR "verticality"[All Fields] OR "vertically"[All Fields] OR "verticals"[All Fields]) OR ("skyscraper"[All Fields] OR "skyscrapers"[All Fields]) OR "tall"[All Fields])
CINAHL:
( ((out-of-hospital cardiac arrest) OR (out of hospital cardiac arrest) OR (OHCA)) ) AND ( ((high rise) OR (high-rise) OR (height)) ) AND ( ((vertical) OR (skyscraper) OR (tall)) )
5. Citations were added into Table 1 as requested.
6. We apologise for this, and have now enhanced the image quality of Figure 2.
Reviewer 2 Report
This is a well-desinged scoping review on a highly significant topic which is not covered in the modern CPR guidelines - cardiac arrest in high-rise buildings. The manuscript is well-written, covers all possible research data in the field, the table is quite informative.
It is advisable to make more clear recommendations for clinicians and lay-people in the discussion section and abstract, higlight main research gaps and directions in this field. It would be more logical to describe first studies related to ROSC, then studies on hospital-to-discharge survival. Conclusions section should be made. References 18 and27 are not full.
Author Response
Thank you for the positive comments and useful suggestions.
- We have outlined clear recommendations for clinicians and lay persons in the discussion section and abstract and highlighted the main research gaps and directions in this field. "A shift to maximising on-scene treatment time, along with bundling novel pre-hospital interventions, could ameliorate some of these difficulties and improve clinical outcomes for patients."
- A conclusion section has now been added.
- We have rechecked references 18 and 27 as advised.
Round 2
Reviewer 1 Report
It appears that all comments have been appropriately responded to. I have no further comments and recommend publication.